# Competitive Chemical Reaction Kinetic Model of Nucleosome Assembly Using the Histone Variant H2A.Z and H2A In Vitro

**DOI:** 10.3390/ijms242115846

**Published:** 2023-10-31

**Authors:** Hongyu Zhao, Xueqin Shao, Mingxin Guo, Yongqiang Xing, Jingyan Wang, Liaofu Luo, Lu Cai

**Affiliations:** 1School of Life Science and Technology, Inner Mongolia University of Science and Technology, Baotou 014010, China; zhaohongyu2000@163.com (H.Z.); sxq1996312@163.com (X.S.); ql3024@139.com (M.G.); xingyongqiang1984@163.com (Y.X.); wangjingyan79@163.com (J.W.); lolfcm@imu.edu.cn (L.L.); 2Inner Mongolia Key Laboratory of Functional Genome Bioinformatics, Inner Mongolia University of Science and Technology, Baotou 014010, China

**Keywords:** histone variant, nucleosome positioning, nucleosome reconstitution in vitro, nucleosome dynamics, competitive chemical reaction kinetics

## Abstract

Nucleosomes not only serve as the basic building blocks for eukaryotic chromatin but also regulate many biological processes, such as DNA replication, repair, and recombination. To modulate gene expression in vivo, the histone variant H2A.Z can be dynamically incorporated into the nucleosome. However, the assembly dynamics of H2A.Z-containing nucleosomes remain elusive. Here, we demonstrate that our previous chemical kinetic model for nucleosome assembly can be extended to H2A.Z-containing nucleosome assembly processes. The efficiency of H2A.Z-containing nucleosome assembly, like that of canonical nucleosome assembly, was also positively correlated with the total histone octamer concentration, reaction rate constant, and reaction time. We expanded the kinetic model to represent the competitive dynamics of H2A and H2A.Z in nucleosome assembly, thus providing a novel method through which to assess the competitive ability of histones to assemble nucleosomes. Based on this model, we confirmed that histone H2A has a higher competitive ability to assemble nucleosomes in vitro than histone H2A.Z. Our competitive kinetic model and experimental results also confirmed that in vitro H2A.Z-containing nucleosome assembly is governed by chemical kinetic principles.

## 1. Introduction

The nucleosome, the basic unit of eukaryotic chromatin, consists of 146 base pairs of DNA wrapped around an octamer of histone proteins containing two copies each of H2A, H2B, H3, and H4. The crystal structure of the nucleosome core particle shows that the DNA wraps around the octamer in about 1.65 superhelix turns, in a left-handed manner and with periodic interaction with histones [1]. The nucleosome structure not only serves as a building block for chromatin to pack DNA but also plays a critical role in the regulation of genome function. As DNA templates, nucleosomes can dynamically regulate all related biological processes, such as transcription, as well as DNA replication, repair, and recombination [2,3].

In addition to the canonical histones, all eukaryotes possess histone variants—in particular, H2A and H3 [4]. H2A.Z, a variant of the H2A histone family, is present in almost all eukaryotic organisms and represents around 15% of the total cellular H2A content, but only 65% of its sequence is conserved between *Saccharomyces cerevisiae* and humans [5]. H2A.Z is encoded by separated genes, such as HTZ1 in *Saccharomyces cerevisiae*, and H2A.Z-1 and H2A.Z-2, located on chromosomes 3 and 11, in mammals. The histone variants are distinguished by an extended acidic patch on the surface of the nucleosome, as well as a unique C-terminal tail, and H2A.Z-1 and H2A.Z-2 differ only by three amino acids in humans [6]. In mammals, H2A.Z is an essential protein [7] that has been shown to play crucial roles in the regulation of gene expression, DNA repair, and chromosome segregation, as well as in the maintenance of heterochromatin–euchromatin boundaries [5,8,9].

Based on a comparison of the canonical histones incorporated into nucleosomes during DNA replication, the histone variant H2A.Z can typically be inserted throughout the cell cycle [10,11,12]. In vertebrates, genome-wide studies have shown that H2A.Z is enriched in active promoters [13,14], facultative heterochromatin, and centromeres [7]. H2A.Z is enriched at the +1 and −1 nucleosome positions surrounding the transcription start site (TSS) across eukaryotes, including mammals, as well as in regulatory regions, such as enhancers and boundary elements [15,16,17]. The H2A.Z fraction in these regions is correlated with the strength of transcription [14,18]. Chromatin remodeling complexes and chaperones may mediate H2A.Z-dependent nucleosome depletion processes in vivo, such as the nucleosome disassembly/assembly chaperone protein Nap1l1, the SWI/SNF complex component Smarca4, the SWR1 component Kat5, and the SWR-C remodeling complex [12,19]. Although the deposition of histone H2A.Z in chromatin is also dependent on chromatin remodelers and histone chaperones, the dynamics of the interaction between histone H2A.Z and canonical histones in nucleosome formation are crucial for understanding the assembly mechanisms of nucleosomes containing H2A.Z. However, the intrinsic dynamics of the H2A.Z-containing nucleosome assembly remain elusive.

In our previous report, we presented a kinetic model of nucleosome assembly in vitro, which confirmed that in vitro nucleosome assembly is governed by chemical kinetic principles [2]. In this model, the efficiency of nucleosome assembly was positively correlated with the total concentration of histone octamers, the reaction rate constant, and the reaction time. All of the corollaries of the model were successfully verified in the canonical nucleosome assembly system in vitro. Although some experimental studies have clearly analyzed the nucleosome crystal structure and described the nucleosome assembly/depolymerization process [20,21], there have been few studies that systematically proposed reaction equations and mathematical models of reaction efficiency with the intent to illuminate the kinetic mechanism from the perspective of chemical reaction kinetic principles. The dynamics model of nucleosome assembly only involving canonical histones and DNA has elucidated the dynamics principle of nucleosome assembly/disassembly in vitro. Here, we first investigated whether this model can also be used to describe the dynamics of H2A.Z-containing nucleosome assembly in vitro, which will further improve the dynamics model of nucleosome assembly/disassembly. In view of the possible competitive relationship between H2A.Z and H2A during nucleosome assembly, a competitive chemical reaction kinetic model of nucleosome assembly, using the histone variant H2A.Z and histone H2A, was also proposed, based on our previous chemical kinetic model of nucleosome assembly. With the proposed model, the competitive ability of histones to assemble nucleosomes can be mathematically described. This model is a very good complement to the knowledge on the assembly dynamics of histone variants and is expected to provide a theoretical basis for further studies on nucleosome assembly dynamics using the histone variant H2A.Z.

## 2. Results

### 2.1. Preparation of DNA Sequences and Histone Octamers

To reconstitute the nucleosome structure in vitro, we first prepared DNA sequences and histone octamers. As shown in Figure 1A–C, the Widom 601 DNA sequence [22] and the CS sequences [23] were used in the nucleosome assembly assays. These DNA sequences were labeled with the fluorescence molecular probe Cy3.

Five recombinant histone proteins (H2A, H2A.Z, H2B, H3, and H4), which have no post-transcriptional modifications, were extracted from bacteria. The canonical histone octamer (Figure 1D) and the H2A.Z-containing octamer (Figure 1E) were reconstituted in refolding buffer and purified using a Superdex S200 filtration column (GE Healthcare Life Science, Uppsala, Sweden). These histone octamers were used to reconstitute nucleosomes.

### 2.2. The H2A.Z-Containing Nucleosome Assembly Can Be Described by a Chemical Kinetic Model

In our previous work [2], we proposed a chemical kinetic model that provides a good description of canonical nucleosome reconstitution in vitro. Based on the chemical reaction kinetic principle, we obtained
(1)$$\frac{1}{Q-S+\gamma \frac{Q+S}{Q-S}}ln\left(\frac{\left\{N-Q\left(1+\frac{\gamma }{Q-S}\right)\right\}\left\{S\left(1-\frac{\gamma }{Q-S}\right)\right\}}{\left\{Q\left(1+\frac{\gamma }{Q-S}\right)\right\}\left\{N-S\left(1-\frac{\gamma }{Q-S}\right)\right\}}\right)=\theta (T)=\bar{k}(\alpha +\varepsilon T)$$
which provides a rigorous expression of *N*/*S* depending on the total concentration of the DNA and histone octamer, reaction rate constant, and reaction time, where *γ* is the integral median of (*k*′/*k*). *N*, *Q*, and *S* indicate the concentration of nucleosomes, total proteins, and total DNA in the reaction system, respectively. $\bar{k}$ and *T* represent the mean reaction constant and total dialysis reaction time.

As *γ*/|*Q* – *S*| ≪ 1, Equation (1) can lead to Equation (2).
(2)$$\frac{N}{S}=\frac{\frac{Q}{S}\left\{1-exp\left[\left(Q-S)\theta (T\right)\right]\right\}}{1-\frac{Q}{S}exp\left\{\left(Q-S)\theta (T\right)\right\}}$$

If $\left(Q-S\right)\theta \left(T\right)<1$, the exponential function in Equation (2) is expanded to the 2nd order of (Q−S)θ(T), and a simplified form of Equation (2) for not-too-large values of *Q* is the linear relationship between *N*/*S* and *Q*:(3)$$\frac{N}{S}=Q\bar{k}(\alpha +\varepsilon T)$$
which is useful in analyzing experimental data.

Equation (3) provides a simplified linear relationship between the nucleosome assembly efficiency and histone concentration, reaction rate constant, and reaction time for the analyses of the experimental data. In Equation (3), *T* represents the total dialysis reaction time, which can be controlled by the dialysis times of the first step. The parameters *N* and *S* denote nucleosome DNA and total DNA, respectively, which can be detected by the fluorescence signal of the Cy3-labeled DNA. *N*/*S* can be used to assess nucleosome assembly efficiency. Next, we tested whether the kinetic model can be used to describe H2A.Z-containing nucleosome reconstitution.

Firstly, we reconstituted H2A.Z-containing nucleosomes on the Widom 601 sequence with a histone octamer concentration gradient to test the relationship between *N*/*S* and *Q*. As shown in Figure 2A, the nucleosome DNA showed an increasing trend as the histone octamer gradient increased. After quantitative analysis of the DNA using gel electrophoresis, we determined the ratio of nucleosome DNA to total DNA as an assembly efficiency metric to evaluate the nucleosome formation ability of each assembled sample. There was a linear relationship between *N*/*S* and *Q* (Figure 2B), which is consistent with Equation (3).

Next, we examined the dependence of the nucleosome assembly efficiency on the reaction time. The H2A.Z-containing nucleosomes were reconstituted at dialysis times of 10, 12, 14, and 16 h by controlling the dilution rate of the dialysis buffer. For the four reaction conditions of 10, 12, 14, and 16 h, the linear slopes between *N*/*S* and *Q* were 0.01361, 0.01384, 0.01442, and 0.01461, respectively (Figure 2B). The result of linear fitting between the slope and reaction time showed that they had a high degree of linear correlation (Figure 2C).

### 2.3. The Characteristics of the DNA Sequence Can Regulate the Assembly Efficiency of H2A.Z-Containing Nucleosomes In Vitro

In the kinetic model, the average reaction rate constant $\bar{k}$ can be used to evaluate the affinity of DNA to histones. Similar to the canonical nucleosome assembly, we also reconstituted H2A.Z-containing nucleosomes on the CS1–CS6 DNA templates to test the application of the constant $\bar{k}$.

CS1 to CS6 are six artificially designed DNA sequences with different sequence features to test the efficiency of nucleosome assembly in vitro. The length of the CS1–CS6 DNA sequences is 156 base pairs. As described in our previous report [23], CS1, CS2, and CS3 fitted the nucleosome positioning pattern RRRRRYYYYY (R5Y5, R = purine and Y = pyrimidine) and contained eleven uninterrupted units of R5Y5. For the control group, the CS4, CS5, and CS6 sequences consisted of eleven uninterrupted copies of the RRYRRYYRYY, RRYYRYYRRY, and RYRYRYRYRY motifs, respectively. In addition, contrary to the CS1 sequence, CS2, CS3, CS4, CS5, and CS6 were characterized by a visible 10.5-base periodicity of TA dinucleotides.

Our previous report indicated that the CS2–CS6 sequences with 10.5 bp TA periodicity had a higher affinity than the CS1 sequence without the 10.5 bp TA periodicity [23]. Based on the present model for the reconstitution of H2A.Z-containing nucleosomes, the slopes of the linear fit of *N*/*S* and *Q* for the CS2–CS6 DNA sequences were significantly higher than that for the CS1 sequence (Figure 3). This result also suggests that the 10.5 bp TA periodicity is a facilitating motif for the assembly of H2A.Z-containing nucleosomes. However, the slope of the linear fit of the CS2 and CS3 sequences with the R5Y5 motif was not significantly higher than that of the CS4 to CS6 sequences without the R5Y5 motif, which is consistent with our previous work [23]. 

As was described in canonical nucleosome assembly, the parameter $\bar{k}$ in the present chemical kinetics model can also be used to evaluate the affinity of DNA fragments to histone H2A.Z-containing octamers, suggesting that the assembly efficiency of H2A.Z-containing nucleosomes in vitro can also be regulated by DNA sequence characteristics.

### 2.4. The Present Model Can Be Used to Describe the Competitive Dynamics of H2A and H2A.Z in Nucleosome Assembly

The competitive chemical kinetic model was verified for in vitro experiments on H2A.Z-containing and canonical nucleosome assemblies. We further investigated whether this model can be used to describe the competitive dynamics of H2A and H2A.Z in nucleosome assembly. 

For H2A and H2A.Z, Equation (2) can be written as
(4)$$\frac{N_{A}}{S}≅Q_{A}{\bar{k}}_{A}(\alpha +\varepsilon T)$$
(5)$$\frac{N_{Z}}{S}≅Q_{Z}{\bar{k}}_{z}(\alpha +\varepsilon T)$$
where subscript characters *A* and *Z* denote H2A and H2A.Z, respectively.

Equations (4) and (5) can be merged into
(6)$$\frac{N_{A}}{N_{Z}}≅\frac{Q_{A}}{Q_{Z}}\times \frac{{\bar{k}}_{A}}{{\bar{k}}_{Z}}$$

In the competitive reaction system, the total concentration of H2A-containing and H2A.Z-containing histone octamers can be controlled quantitatively. In the reconstituted nucleosome sample, H2A-containing and H2A.Z-containing nucleosomes can be directly detected by a combination of native PAGE and Western blotting. By linear fitting between $\frac{Q_{A}}{Q_{Z}}$ and $\frac{N_{A}}{N_{Z}}$, $\frac{{\bar{k}}_{A}}{{\bar{k}}_{Z}}$ can be obtained. If $\frac{{\bar{k}}_{A}}{{\bar{k}}_{Z}}$ < 1, the competitive ability of histone H2A.Z to assemble nucleosomes is stronger than that of histone H2A; otherwise, the ability of H2A is stronger than H2A.Z in the case of $\frac{{\bar{k}}_{A}}{{\bar{k}}_{Z}}$ > 1.

We then examined the competitive abilities of H2A and H2A.Z in nucleosome assembly using a competitive reaction system. In the reaction system, the total concentration of histones was constant, and a gradient of histone H2A or H2A.Z concentration was used. Nucleosome DNA and free DNA were separated from the reconstituted nucleosome sample by native PAGE. The results showed that the nucleosomes could be reconstituted on Widom 601 DNA with each histone mix (Figure 4A). We further employed Western blotting to detect the H2A.Z-containing and canonical nucleosome contents in the nucleosome mixture sample by transferring the nucleosomes from the polyacrylamide gel to a PVDF membrane. As shown in Figure 4B, H2A.Z and H2A were identified in nucleosome samples. To reduce the effects of experimental errors, the protein levels quantified using the ImageJ software (1.53q) were standardized based on the maximum value. Through the linear fitting between $\frac{N_{A}}{N_{Z}}$ and $\frac{Q_{A}}{Q_{Z}}$, we obtained $\frac{{\bar{k}}_{A}}{{\bar{k}}_{Z}}$ = 1.71 (Figure 4C). When the total histone mass increased to 3.6 μg using the same reaction system with 3 μg DNA, $\frac{{\bar{k}}_{A}}{{\bar{k}}_{Z}}$ = 1.59, and when the total histone mass decreased to 2.4 μg, $\frac{{\bar{k}}_{A}}{{\bar{k}}_{Z}}$ = 1.38. These results suggest that histone H2A has a stronger competitive ability to assemble nucleosomes than histone H2A.Z in vitro.

Subsequently, we compared the dissociation dynamics of canonical and H2A.Z-containing nucleosomes at high temperatures. The thermal stability assay was employed to detect the thermal stability dependence of nucleosome disassembly. The nucleosome was reconstituted on a 147 bp 601 DNA template without any fluorescence labeling. The thermal stability assay monitors the fluorescence signal from SYPRO Orange, which binds hydrophobically to the proteins through thermal denaturation. In this assay, the histones that thermally dissociate from the nucleosome are detected by the fluorescent signal of SYPRO Orange [2,24]. As shown in Figure 5A, the fluorescence signal intensity began to increase significantly at 55 °C, which indicated that the nucleosomes started to decompose and the hydrophobic domains in the histones were exposed. The canonical nucleosomes had an obvious rapid increase in the fluorescence in the first stage from 68 to 75 °C compared to the H2A.Z-containing nucleosomes, which indicated that the removal of the H2A/H2B dimers from the canonical nucleosomes was easier than that of H2A.Z/H2B dimers from the H2A.Z-containing nucleosomes. In the later stage from 76 to 87 °C, H3/H4 tetramers dissociated from the DNA. Then, we examined the disassembly rate of nucleosomes under the incubation condition of 75 °C. The proportions of nucleosome DNA and free DNA, which were separated in the gel and quantified by Cy3 labeling, were used to evaluate the rate of nucleosome disassembly. The results in Figure 5B indicate that the canonical nucleosomes disassembled much faster than the H2A.Z-containing nucleosomes. These results also support the hypothesis that H2A has a higher reaction rate constant than H2A.Z in nucleosome reconstitution.

## 3. Discussion

In the present study, the dynamics of the integration of the histone variant H2A.Z into nucleosomes were investigated in vitro based on a previous chemical kinetic model of canonical nucleosome assembly. The results showed that the nucleosome assembly efficiency of the H2A.Z-containing nucleosomes was positively correlated with the average reaction rate constant, histone octamer concentration, and reaction time. The experimental validation of the kinetic model indicates that the in vitro assembly of H2A.Z-containing nucleosomes is also governed by chemical kinetic principles.

As an evolutionarily conserved histone variant, H2A.Z has been found to have many functions in gene transcription, DNA repair, DNA replication, and chromatin structure, such as in the pausing and elongation of RNA polymerase II (RNAPII), enhancer activity, active and inactive gene transcription [13,25,26], nucleosome turnover [27], heterochromatin boundaries [28,29], and chromosome segregation [30,31]. The histone H2A family encompasses the largest number of variants [32], with H2A.Z variants representing approximately 15% of the total cellular H2A content [4]. A high-resolution map of H2A.Z ChIP-seq data showed the formation of two well-positioned H2A.Z nucleosomes flanking each side of the TSS (the −1 and +1 nucleosomes) in humans and mice. It has also been reported that an unstable H2A.Z-containing nucleosome occupies the TSS of an active promoter [33,34,35]. Despite considerable efforts to understand the deposition and removal dynamics of H2A.Z on eukaryotic chromatin, its deposition and removal mechanisms remain enigmatic because the in vivo process is instantaneous and complex. 

The role of H2A.Z in disease has been extensively studied, particularly in cancer [31]. Mutations, transcriptional deregulation, and changes in the deposition machineries of histone variant H2A.Z can affect the process of tumorigenesis. These alterations promote or even drive cancer development through mechanisms that involve changes in epigenetic plasticity, genomic stability, and senescence and by activating and sustaining cancer-promoting gene expression programs [36]. The regulatory mechanisms of H2A.Z in diseases can be roughly summarized in two ways. The overexpression of H2A.Z in different types of diseases, especially in cancers, is one important regulation pathway. The change in the chromatin structure, recruitment of transcription factors, and coordination of other histone modifications are involved in the abnormal gene expression caused by H2A.Z overexpression in diseases. H2A.Z is upregulated and contributes to the disease-associated gene activation that drives proliferation, epithelial–mesenchymal transition (EMT), metastasis, and cancer in many diseases, such as hormone-dependent breast cancer [36], prostate cancer [37], liver cancer [38], hepatocellular carcinoma [38], melanoma [39], and uterine leiomyoma [40]. H2A.Z silencing can inhibit the proliferation and invasion of intrahepatic cholangiocarcinoma (ICC) cells [41]. On the other hand, the deposition of H2A.Z into the nucleosome or eviction from chromatin is dependent on chromatin remodelers or histone chaperones. The deregulation of histone chaperone complexes can contribute to tumor development, and both chaperone complexes, SRCAP and p400–TIP60, are affected. Components of the SRCAP complex are upregulated in several tumors, such as ovarian tumors, breast tumors, thyroid tumors, prostate tumors, melanoma, bladder tumors, glioma, and Floating-Harbor syndrome (FHS) [42,43]. Furthermore, the chromatin remodeler p400 is overexpressed in some cancers [38] and promotes oncogenic WNT signaling and counteracts the tumor-suppressive function of the histone acetyltransferase KAT5 [44]. An aberrant nucleosome assembly occurs, while excessive H2A.Z deposition in the genome is mediated by H2A.Z overexpression and chromatin remodelers in various diseases. The assembly dynamics of nucleosomes containing H2A.Z, especially the competitive dynamics of H2A.Z and H2A in nucleosome assembly, have an important implication for the role of H2A.Z in disease.

Our model describes the assembly dynamics of H2A.Z-containing nucleosomes based on in vitro nucleosome assembly and disassembly assays. In vivo nucleosome reconstitution, dissociation, and remodeling, especially the competitive assembly of canonical histones and histone variants, are more complicated than those in vitro. The deposition of H2A.Z into the nucleosome or eviction from chromatin is dependent on chromatin remodelers or histone chaperones in vivo. For example, the SWR1, p400/Tip60, and SRCAP complexes can load H2A.Z onto chromatin, and ANP32E and INO80 complexes can remove H2A.Z from the nucleosome in human cells during DNA damage [31,42,45,46,47]. These chromatin remodeling enzymes use the energy of ATP hydrolysis to catalyze a histone exchange event in which each of the two nucleosomal H2A–H2B dimers is sequentially replaced by an H2A.Z–H2B variant dimer [48]. Our kinetics model of H2A.Z-containing nucleosome assembly may not be able to directly describe the kinetics of nucleosome assembly dynamics in vivo. However, it does provide a model of the intrinsic kinetics of the H2A.Z-containing nucleosome assembly, which only involves the interaction of DNAs and histones, and could be used to elucidate the basic kinetic principles of nucleosome assembly. On the other hand, the applicability of this kinetics model to H2A.Z-containing nucleosome assembly indicated that chemical kinetics is a basic principle for all nucleosome assembly in vitro, which could also provide an ideal model to develop a kinetic model of nucleosome assembly [2].

In vivo, DNA wraps around histone octamers to form nucleosomal structures. In this process, the DNA needs to overcome a large bending force to contact the protein surface due to the interaction between DNA and protein. There are obvious characteristic differences such as bending, rotation, and deformation among base pairs of DNA sequences. Therefore, DNA sequence characteristics must affect the assembly efficiency of nucleosomes. The assembly process of H2A.Z-containing nucleosomes in vivo is regulated by chromatin remodelers or histone assembly chaperones. The effect of DNA sequences on the assembly of H2A.Z nucleosomes is not as large as that on canonical nucleosomes, but our experiments still demonstrate that some DNA sequence features that regulate nucleosome positioning can affect the assembly efficiency of H2A.Z nucleosomes in the absence of any nucleosome assembly regulator.

In eukaryotic genomes, the canonical nucleosome is the dominant nucleosome, and histone variants are in the minority. Under in vitro conditions, our experiments confirmed that canonical nucleosomes have a stronger assembly capacity than H2A.Z-containing nucleosomes, which is also consistent with the in vivo nucleosome abundance. Therefore, without the presence of histone chaperones or nucleosome assembly factors for the special regulatory requirements, chromatin still tends to be occupied by canonical nucleosomes due to the high reaction rate constant for canonical nucleosome assembly.

The mean reaction rate constant is an important parameter in the model. In the nucleosome assembly process through salt dialysis, a decrease in salt concentration provides an impetus for nucleosome assembly. The average reaction rate constant can be considered an indicator of the rate of the reaction process for different DNA templates, and it can further assess the affinity of the DNA for the histone H2A.Z-containing octamer. For the six CS sequences, the experimental results for their affinity to the H2A.Z-containing octamer are consistent with those in our previous work under certain nucleosome assembly conditions [23]. The average reaction rate constant in our chemical kinetic model provides a comprehensive parameter to describe the nucleosome assembly rate and can further evaluate the formation ability of H2A.Z-containing nucleosomes. All of the experimental results indicate that the H2A.Z-containing nucleosome assembly in vitro obeys our kinetic model and is governed by chemical kinetic principles.

The present model still has some limitations. Nucleosome positioning along the genome might be determined primarily by intrinsic DNA sequence preferences for histone and external factors, such as chromatin remodeling, DNA methylation, histone variants, post-translational modifications, and elongation by polymerase II [23]. However, only DNA and histones were considered in the nucleosome assembly model. In future investigations, histone chaperones and chromatin remodelers should be integrated into the model for a more comprehensive mechanism of nucleosome assembly. This kinetic model has been expanded to include the competitive dynamics of H2A and H2A.Z in nucleosome assembly, which has provided a novel method for assessing the competitive ability of histones to assemble nucleosomes. Next, we will add RNA polymerase II into the nucleosome assembly model and nucleosome assembly reaction system. By analyzing the competitive binding of RNA polymerase II and histones to DNA, we can attempt to understand the coupling mechanism of transcription elongation and nucleosome dynamics. We will also develop the model to investigate the effect of characteristics of DNA, histone variables, and physiological variables on competitive nucleosome assembly with canonical histones and histone variants. In addition, a complex in vitro system for nucleosome reconstitution can be constructed by combining salt dialysis, histone chaperones, and ATP-dependent assembly factors. This nucleosome reconstitution system can be used to examine more complicated factors in a theoretical model. The new model may better reflect the in vivo nucleosome dynamics.

In conclusion, our previously proposed chemical kinetic model for nucleosome assembly can also be applied to the assembly processes of H2A.Z-containing nucleosomes. We have further extended this model to describe the competitive dynamics of H2A and H2A.Z in nucleosome assembly, which provides a novel way to evaluate the competitive ability of histones to assemble nucleosomes. Based on this model, we confirmed that histone H2A has a stronger competitive ability to assemble nucleosomes in vitro than histone H2A.Z. These results will contribute to a deeper understanding of the in vivo dynamics of nucleosome assembly. However, in vivo nucleosome assembly/disassembly is more complicated and involves histone variants, chromatin remodelers, RNA polymerase II, and many other factors. In the future, these complex factors will be integrated into the dynamics model for to help further understand the cooperative mechanism of nucleosome assembly/disassembly in vivo, which will lay a basis for understanding the mechanism of nucleosome structures in gene expression regulation.

## 4. Materials and Methods

### 4.1. Preparation of DNA Sequences and Recombinant Histone Octamers

The Widom 601 DNA sequence [22] and CS sequences [23] were used to assemble H2A.Z-containing nucleosomes in vitro. For the quantitative analysis of the DNA bands in the gel, the DNA was labeled with Cy3 by PCR. For a PCR of 147 bp 601 DNA, the forward primer used was 5′-Cy3-CAGGATGTATATATCTGACACGTGCCT-3′ and the reverse primer was 5′-CTGGAGAATCCCGGTGCCGAGGCC-3′. For the six artificially synthesized CS1–CS6 DNA sequences, the detailed sequence information was shown in our previous paper [23]. The length of the CS1–CS6 DNAs is 156 base pairs. The forward PCR primer was 5′-Cy3-ACGGCCAGTGAATTCGAGG-3′, and the reverse primer was 5′-GCCAAGCTTCTGAGATCGGAT-3′.

The expression and purification of the histones were performed as previously described [2,23,49]. Briefly, five histones (H2A, H2A.Z, H2B, H3, and H4) were expressed and purified from *E. coli* BL21 cells containing pET-histone expression plasmids. To reconstitute canonical or H2A.Z-containing histone octamers, the four histones were mixed at equimolar ratios in an unfolding buffer (7 M guanidinium HCl; 20 mM Tris-HCl, pH 7.5; and 10 mM DTT). The mixture was dialyzed in a refolding buffer (2 M NaCl; 10 mM Tris-HCl, pH 7.5; 1 mM Na-EDTA; and 5 mM 2-mercaptoethanol) at 4 °C. After removal of the precipitated proteins by centrifugation for 30 min at 20,000× *g*, the samples were concentrated to a final volume of 0.5 mL and purified through a Superdex S200 filtration column (GE Healthcare Life Science, Uppsala, Sweden). Confirmation of the purity and stoichiometry of the histone octamers was performed using SDS-PAGE on a 15% gel stained with Coomassie Brilliant Blue, and the concentration was determined using an extinction coefficient at 276 nm.

### 4.2. Nucleosome Assembly Reaction In Vitro

The salt dialysis method was used to reconstitute H2A.Z-containing nucleosomes as described previously [2,23]. Generally, each DNA fragment was incubated in a reconstitution reaction containing 10 mM Tris-HCl (pH 8.0), 1 mM EDTA (pH 8.0), 2 M NaCl, and histone octamers. The samples were dialyzed from a high-salt buffer (2 M NaCl; 10 mM Tris-HCl, pH 8.0; and 1 mM EDTA) to a lower concentration of 0.6 M NaCl for 16 h. The samples were then dialyzed in a TE buffer (10 mM Tris-HCl, pH 8.0; 1 mM EDTA) for an additional minimum of 3 h and then used for further gel analysis. Dialysis was performed in a darkroom at 4 °C for the assembly reaction on fluorescence-labeled DNA templates. For the time-course nucleosome assembly experiments, the duration of the first-step dialysis with the buffer containing 2 M to 0.6 M salt was set at 10, 12, 14, and 16 h. The second dialysis was performed for an additional minimum of 3 h in TE buffer without NaCl for all assembly experiments.

In the reaction system, 3 μg of DNA template in a 60 μL total reaction volume was used to assemble nucleosomes. As described in our previous work [2], the molar concentration of 601 DNA sequences was 5.09 × 10^−7^ mol/L, and the molar concentration of the CS DNA sequences was 4.62 × 10^−7^ mol/L in the reaction system. The mass concentrations of the histone octamers in the reaction system were tested in a gradient from 5 μg/mL to 60 μg/mL, and the molar concentrations ranged from 0.46 × 10^−7^ mol/L to 5.52 × 10^−7^ mol/L.

For the H2A.Z and H2A competing reactions, 2.4 μg, 3 μg, and 3.6 μg of total histone proteins containing canonical and H2A.Z-containing octamers in three nucleosome reconstitution reactions were used to reconstitute nucleosomes on 3 μg of DNA template. As shown in Table 1, six different concentrations of H2A and H2A.Z histones were tested in each reaction system.

### 4.3. Gel Analysis of Nucleosome Assembly Efficiency

All of the DNA templates were labeled with the fluorescent molecule Cy3. The reaction mixtures were resolved on 5% native polyacrylamide gels in 0.5 × TBE. Cy3 fluorescence was measured and quantified in the nucleosome DNA and the free DNA bands in the gel at an emission wavelength of 605 nm and an excitation wavelength of 520 nm (GE Healthcare, Amersham Imager 600RGB (GE Healthcare Life Science, Tokyo, Japan) and Image Quant TL, v8.1.0.0).

After quantitative analysis of the DNA in the gel, the ratio of nucleosome DNA to total DNA was quantified as the assembly efficiency to evaluate the nucleosome formation ability of each assembled sample. The linear relationship between *N*/*S* and *Q* was fitted for the following analyses using Equation (3), where the nucleosome assembly efficiency of 0 μg/mL of histone octamers was set as 0 to obtain slopes.

### 4.4. Native PAGE/Western Blotting Analysis of the Competitive Ability of the H2A and H2A.Z Histones in Reconstituted Nucleosomes

Native PAGE/Western blotting was employed to detect the efficiency of nucleosome assembly with the H2A and H2A.Z histones. As described in Section 4.2, the mixed nucleosomes were reconstituted using canonical histone octamers and H2A.Z-containing histone octamers. After confirmation of successful nucleosome reconstitution, antibodies against H2A, H2A.Z, and H4 were used to detect the specific histones in the reconstituted nucleosome.

Firstly, native PAGE separation of the samples from the H2A.Z and H2A competing nucleosome assembly systems was used to confirm the reconstituted nucleosomes. The Cy3 fluorescence of nucleosome DNA and the free DNA bands was observed at an emission wavelength of 605 nm and an excitation wavelength of 520 nm.

Three similar native PAGE experiments were performed for the detection of H2A, H2A.Z, and H4 in the nucleosomes. The reconstituted samples containing equal masses of proteins and DNA were loaded for electrophoresis analysis. After native PAGE separation, the histones in the nucleosome were transferred to polyvinylidene difluoride (PVDF) membranes. The membranes were blocked with 5% skim milk at room temperature in a TBST buffer, and the indicated antibodies (H2A: Abcam, Waltham, MA, USA, ab18255; H2A.Z: Abcam, ab4174; H4: Abcam, ab31830) were used. The immunoreactive bands were visualized using horseradish–peroxidase-conjugated secondary antibodies and an enhanced chemiluminescent substrate. Protein levels were quantified using Image J software (1.53q). To reduce the effects of experimental errors, the band intensity of the maximum input of histone H2A or H2A.Z in the reaction system was defined as 1, and the protein levels were standardized based on the maximum value in each Western blotting assay. Through linear fitting of $\frac{N_{A}}{N_{Z}}$ vs. $\frac{Q_{A}}{Q_{Z}}$ and setting the vertical intercept to zero, we obtained the value of the slope $\frac{{\bar{k}}_{A}}{{\bar{k}}_{Z}}$.

### 4.5. Thermal Stability Assay

The stability of nucleosome dissociation was evaluated using the previously described thermal stability shift assay [2,24,50]. The canonical and H2A.Z-containing nucleosomes were reconstituted using 147 bp 601 fragments without any labeling. The thermal stability assay was performed in a solution containing 0.25 M NaCl, 10 mM HEPES, 1 mM β-mercaptoethanol, and 5 × SYPRO Orange. The amount of nucleosomes was equivalent to 375 ng of DNA per reaction. The total volume was adjusted to 30 mL. The fluorescence signals of SYPRO Orange were recorded in the VIC channel of a real-time PCR detection system (ABI 7500), and a temperature gradient of 25 to 95 °C was used with a 1 °C step.

The raw fluorescence intensity data were normalized using the formula ${NF}_{i}=\frac{F_{i}-F_{min}}{F_{max}-F_{min}}$, where *F_i_*, *NF_i_*, *F_min_*, and *F_max_* indicate the fluorescence at a certain temperature, normalized fluorescence, and the minimum and maximum fluorescence intensity, respectively. The temperature range was 50–95 °C.

## 5. Conclusions

The chemical kinetic model for nucleosome assembly can be extended to the H2A.Z-containing nucleosome assembly processes. The efficiency of H2A.Z-containing nucleosome assembly was positively correlated with the total histone octamer concentration, reaction rate constant, and reaction time. The proposed kinetic model for describing the competitive dynamics of H2A and H2A.Z in nucleosome assembly provides a novel method to assess the competitive ability of histones to assemble nucleosomes. The in vitro experiments demonstrated that histone H2A exhibits a higher competitive ability to assemble nucleosome than histone H2A.Z. Both the competitive kinetic model and the experimental results indicate that in vitro H2A.Z-containing nucleosome assembly is governed by chemical kinetic principles.

## Figures and Tables

**Figure 1 ijms-24-15846-f001:**
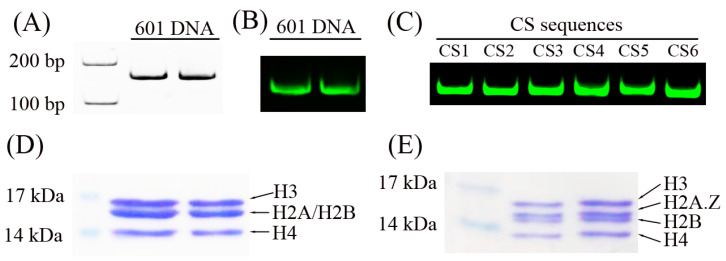
DNA and histone octamers for the nucleosome reconstitution assay. (**A**) Two repeated 601 DNA sequences were detected by native PAGE and ethidium bromide staining. (**B**) Two repeated Cy3-labeled 601 DNA sequences were detected by native PAGE. (**C**) Six different Cy3-labeled DNA sequences, referred to as CS1 to CS6, were detected by native PAGE using the Cy3 fluorescence signal. (**D**) SDS-PAGE analysis of two repeated reconstituted canonical histone octamers. (**E**) SDS-PAGE analysis of two repeated reconstituted H2A.Z-containing histone octamers.

**Figure 2 ijms-24-15846-f002:**
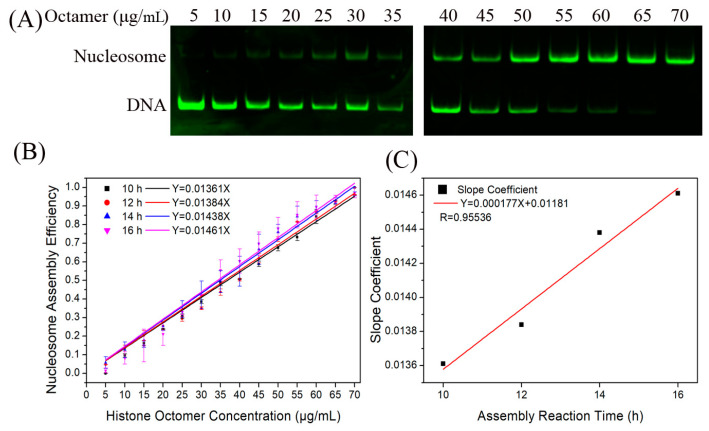
Reconstitution of H2A.Z-containing nucleosomes in vitro. (**A**) Detection results of nucleosome assembly in vitro. The reconstituted nucleosomes with different histone octamers were analyzed by native PAGE. In each lane of the gel, the top band is nucleosome DNA and the bottom band is free DNA. (**B**) Regression curves of nucleosome assembly efficiency vs. histone octamer concentration for 10, 12, 14, and 16 h of first-step dialysis. In accordance with the formulas, we set the nucleosome assembly efficiency at 0 μg/mL of histone octamers to 0 to produce slopes. For each sample, five independent repeats were performed. (**C**) Linear regression curve between the slope coefficient in panel (**B**) and the assembly reaction time.

**Figure 3 ijms-24-15846-f003:**
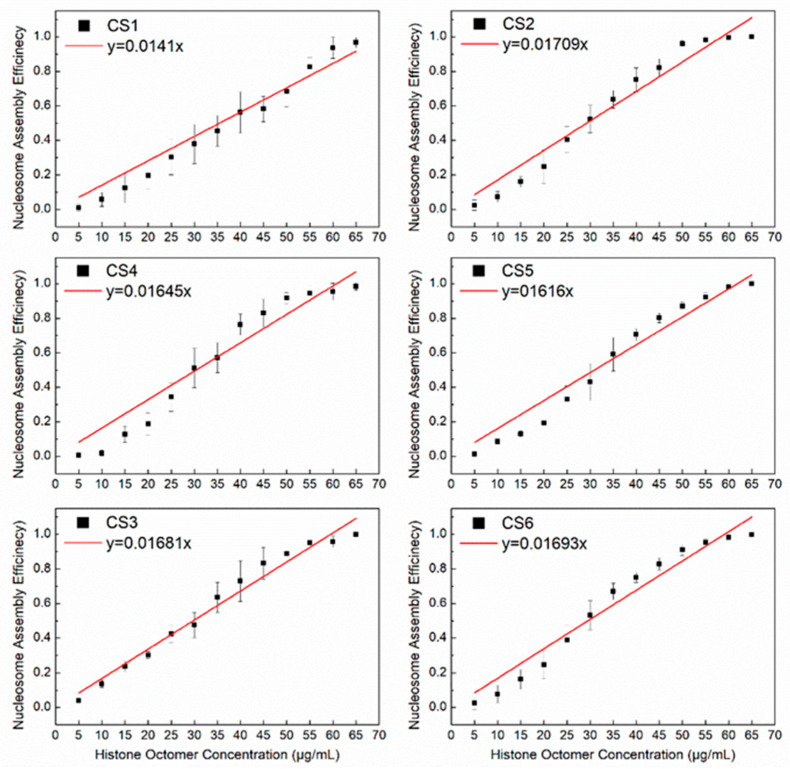
The regression curves of H2A.Z-containing nucleosome assembly efficiency vs. the histone octamer concentration for six CS sequences. The nucleosomes were reconstituted on CS1–CS6 DNA sequences with different histone octamer concentrations. Reconstituted nucleosomes were analyzed by native PAGE and quantized to calculate the nucleosome assembly efficiency. For each sample, five independent repeats were performed.

**Figure 4 ijms-24-15846-f004:**
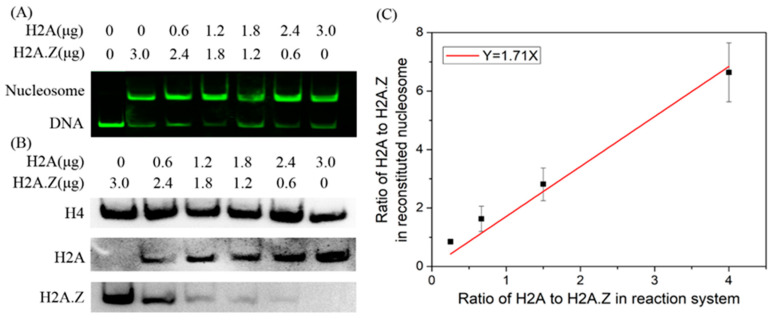
The competitive reaction assay to examine the competitive abilities of H2A and H2A.Z in nucleosome assembly. (**A**) Native PAGE results for competitive nucleosome assembly in vitro. (**B**) Western blot results of histones in reconstituted nucleosomes. (**C**) Linear fitting of the ratio of H2A and H2A.Z in reconstituted nucleosomes to the ratio of H2A and H2A.Z in reaction systems.

**Figure 5 ijms-24-15846-f005:**
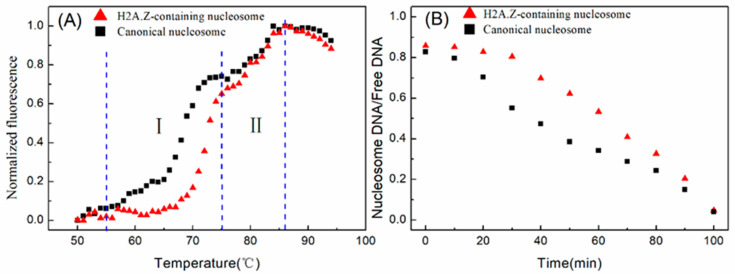
Nucleosome disassembly in vitro under high temperature conditions. (**A**) Thermal shift assays on the nucleosomes using SYPRO Orange. The relative fluorescence intensity at each temperature is plotted as a black dot (canonical nucleosomes) or red dot (H2A.Z-containing nucleosomes). The temperature interval I denotes 55 to 75 °C. The temperature interval II denotes 76 to 87 °C. (**B**) The disassembly of nucleosomes at a temperature of 75 °C.

**Table 1 ijms-24-15846-t001:** The competing reaction system for H2A.Z and H2A nucleosome reconstitution.

Reaction Gradient	2.4 μg Total Proteins		3 μg Total Proteins		3.6 μg Total Proteins	
	H2A	H2A.Z	H2A	H2A.Z	H2A	H2A.Z
Gradient 1	0	2.4	0	3	0	3.6
Gradient 2	0.48	1.92	0.6	2.4	0.72	2.88
Gradient 3	0.96	1.44	1.2	1.8	1.44	2.16
Gradient 4	1.44	0.96	1.8	1.2	2.16	1.44
Gradient 5	1.92	0.48	2.4	0.6	2.88	0.72
Gradient 6	2.4	0	3	0	3.6	0

## Data Availability

All data generated or analyzed during this study are included in the published article.

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
