# Peer review of "Competitive Chemical Reaction Kinetic Model of Nucleosome Assembly Using the Histone Variant H2A.Z and H2A In Vitro"

_ijms, 2023, doi:10.3390/ijms242115846_

Round 1
Reviewer 1 Report
Comments and Suggestions for Authors
Main Contribution and Strengths:
The pivotal contribution of this paper lies in the formulation of a competitive chemical reaction kinetic model, meticulously developed for understanding nucleosome assembly utilizing histone variants H2A.Z and H2A in vitro. The authors delve into a detailed analysis of the assembly dynamics of H2A.Z-containing nucleosomes, juxtaposing it with H2A-containing nucleosomes. Noteworthy strengths of the paper encompass the employment of a synergistic approach, utilizing native-PAGE and Western blotting to directly discern H2A and H2A.Z-containing nucleosomes, coupled with the development of a mathematical model to elucidate the assembly kinetics.
General Comments:
The paper is commendably well-written and organized, presenting a coherent flow of information and analysis. Nonetheless, there are several aspects that warrant enhancement:
-
Experimental Procedures: A more in-depth exposition of the experimental procedures, especially those employed to detect H2A and H2A.Z-containing nucleosomes, would augment the reproducibility and verifiability of the findings.
-
Model Limitations: A discussion delineating the potential limitations of the developed model, accompanied by suggestions for future research trajectories, would bolster the paper’s credibility and relevance.
Review Completeness:
The authors have adeptly covered the review topic, ensuring appropriateness in relation to the paper’s scope. A comprehensive literature review on nucleosome assembly and the role of histone variant H2A.Z is provided. However, the following could be enhanced:
- Knowledge Gap: An elaboration on the identified knowledge gap and a clearer depiction of how the present study endeavors to bridge this gap would enhance the paper’s significance and rationale.
Special Comments:
While the paper is informative and provides valuable insights into the assembly dynamics of H2A.Z-containing nucleosomes, the following considerations could further enrich it:
-
Biological Implications: A more detailed discussion regarding the biological implications of the findings, particularly how the assembly dynamics of H2A.Z-containing nucleosomes diverge from H2A-containing nucleosomes in vivo, would be beneficial.
-
Clinical Relevance: Delving into the potential clinical relevance of the findings, especially considering the implication of aberrant nucleosome assembly in various diseases, would provide a broader context and potential applicability of the research.
Conclusion:
Overall, the paper is well-constructed, informative, and provides valuable insights into nucleosome assembly dynamics. With the incorporation of the aforementioned enhancements, it has the potential to be a significant contribution to the field.
Author Response
International Journal of Molecular Sciences
Competitive chemical reaction kinetic model of nucleosome assembly using histone variant H2A. Z and H2A in vitro
Manuscript ID: ijms-2664452
Authors: Hongyu Zhao, Xueqin Shao, Mingxin Guo, Yongqiang Xing, Jingyan Wang, Liaofu Luo, Lu Cai
Dear reviewers and Ms. Aisa Safaya,
Thanks to the reviewers and editor for your helpful comments on our manuscript. We really appreciate that you give us the opportunity to revise the manuscript. We carefully revised the manuscript with consideration of reviewers’ comments and detailed corrections are listed below point by point.
Besides those changes made according to the reviewers’ suggestions, some minor revisions that will not influence the content and framework of the manuscript were also made in order to submit the manuscript more clearly. We do not list these minor revisions here. All changes in the revised manuscript are highlighted in yellow.
We hope this will make it acceptable for publication.
Sincerely yours,
Lu Cai (Email: nmcailu@163.com)
Oct.18, 2023
Reply to the reviewer 1:
Comment 1: The paper is commendably well-written and organized, presenting a coherent flow of information and analysis. Nonetheless, there are several aspects that warrant enhancement:
Experimental Procedures: A more in-depth exposition of the experimental procedures, especially those employed to detect H2A and H2A.Z-containing nucleosomes, would augment the reproducibility and verifiability of the findings.
Reply: According to reviewer’s suggestion, we gave a more in-depth exposition of the experimental procedures in revised manuscript, especially the detection method of H2A and H2A.Z-containing nucleosomes. The changed details were highlighted with bright yellow in the section 4.2, 4.3, and 4.4 of revised manuscript.
Comment 2: Model Limitations: A discussion delineating the potential limitations of the developed model, accompanied by suggestions for future research trajectories, would bolster the paper’s credibility and relevance.
Reply: According to reviewer’s suggestion, we have added the limitations of the model and suggestions for future research trajectories in the discussion section of revised manuscript.
The present model still has some limitations. Nucleosome positioning along the genome might be determined primarily by intrinsic DNA sequence preferences on histone and external factors, such as chromatin remodeling, DNA methylation, histone variants, post-translational modifications, and elongation of polymerase II. However, the DNA and histone were only considered in the nucleosome assembly model. In the future investigation, histone chaperones, and chromatin remodelers should be integrated into the model for more comprehensively elucidating the mechanism of nucleosome assembly. This kinetic model has been expanded to delineate the competitive dynamics of H2A and H2A.Z in nucleosome assembly, which has provided a novel method to assess the competitive ability of histones to assemble nucleosomes. Next, we will add RNA polymerase Ⅱ into nucleosome assembly model and nucleosome assembly reaction system. By analyzing the competitive binding of RNA polymerase Ⅱ and histones to DNAs, we can attempt to understand the coupling mechanism of transcription elongation and nucleosome dynamics. We will also develop the model to investigate the effect of characteristics of DNAs, histone variables and physiological variables on competitive nucleosome assembly with canonical histones and histone variants. In addition, a complex system of nucleosome reconstitution in vitro can be constructed by combining salt dialysis, histone chaperones and ATP-dependent assembly factors. This nucleosome reconstitution system can be used to examine more complicated factors in theoretical model. The new model may get closer to nucleosome dynamics in vivo.
Comment 3: The authors have adeptly covered the review topic, ensuring appropriateness in relation to the paper’s scope. A comprehensive literature review on nucleosome assembly and the role of histone variant H2A.Z is provided. However, the following could be enhanced:
Knowledge Gap: An elaboration on the identified knowledge gap and a clearer depiction of how the present study endeavors to bridge this gap would enhance the paper’s significance and rationale.
Reply: In the introduction section, we added an elaboration on the identified knowledge gap and a clearer depiction of the present study. The details are showed as follows.
In our previous report, we presented a kinetic model of nucleosome assembly in vitro, which confirm that nucleosome assembly in vitro is governed by chemical kinetic principles. In the model, the efficiency of nucleosome assembly is positively correlated with the total concentration of histone octamer, the reaction rate constant, and the reaction time. All the corollaries of the model are successfully verified in the canonical nucleosome assembly system in vitro. Although some experimental studies have clearly analyzed nucleosome crystal structure and described the nucleosome assembly/depolymerization process, there has been few studies to systematically propose reaction equations and mathematical models of reaction efficiency to illuminate the kinetic mechanism from the perspective of chemical reaction kinetic principles. The dynamics model of nucleosome assembly only in-volved with canonical histone and DNA elucidated the dynamics principle of nucleosome assembly/disassembly in vitro. Here we first investigate whether this model can be also used to describe the dynamics of H2A.Z-containing nucleosome assembly in vitro, which will further improve the dynamics model of nucleosome assembly/disassembly. In view of the possible competitive relationship between H2A.Z and H2A in nucleosome assembly, a competitive chemical reaction kinetic model of nucleosome assembly using the histone variant H2A.Z and histone H2A was further proposed based on our previous chemical kinetic model of nucleosome assembly. In the model, the competitive ability of histones to assemble nucleosomes can be mathematically described. This model is a very good complement to the dynamic knowledge of the present experimental description of histone variants, and is expected to provide a theoretical basis for further understanding of the nucleosome assembly dynamics using the histone variant H2A.Z.
Comment 4: While the paper is informative and provides valuable insights into the assembly dynamics of H2A.Z-containing nucleosomes, the following considerations could further enrich it:
Biological Implications: A more detailed discussion regarding the biological implications of the findings, particularly how the assembly dynamics of H2A.Z-containing nucleosomes diverge from H2A-containing nucleosomes in vivo, would be beneficial.
Reply: We have added a more detailed discussion regarding the biological implications of the findings in revised manuscript.
Our model describes the assembly dynamics of H2A.Z-containing nucleosomes based on nucleosome assembly and disassembly assays in vitro. Nucleosome reconstitution, dissociation and remodeling in vivo, especially competitive assembly of canonical histones and histone variants, are more complicated than that in vitro. The deposition of H2A.Z into the nucleosome or eviction from chromatin is more dependent on chromatin remodelers or histone chaperones in vivo. For example, the SWR1, p400/Tip60, and SRCAP complexes can load H2A.Z within chromatin, and ANP32E and INO80 complexes can remove H2A.Z from the nucleosome in human cells during DNA damage. These chromatin remodeling enzymes use the energy of ATP hydrolysis to catalyze a histone exchange event in which each of the two nucleosomal H2A-H2B dimers is sequentially replaced by an H2A.Z-H2B variant dimer. Our kinetics model of H2A.Z-containing nucleosome assembly may not be directly used to describe the apparent kinetics of nucleosome dynamics in vivo. However, the model provided an intrinsic kinetics of H2A.Z-containing nucleosome assembly, which only involve the interaction of DNAs and histones, may elucidate the basic rule in the kinetic principle of nucleosome assembly and completion process. On the other hand, the well applying of this kinetics model to H2A.Z-containing nucleosome assembly indicated that the chemical kinetics is a basic principle for all nucleosome assembly in vitro, which could also provide the ideal model to develop further an apparent kinetics model of nucleosomes.
Comment 5: Clinical Relevance: Delving into the potential clinical relevance of the findings, especially considering the implication of aberrant nucleosome assembly in various diseases, would provide a broader context and potential applicability of the research.
Reply: As the following description, the clinical relevance has been added in the discussion section of revised manuscript.
The role of H2A.Z in disease has been extensively studied, particularly in cancer. Mutations, transcriptional deregulation and changes in the deposition machineries of histone variant H2A.Z can affect the process of tumorigenesis. These alterations promote or even drive cancer development through mechanisms that involve changes in epigenetic plasticity, genomic stability and senescence, and by activating and sustaining cancer-promoting gene expression programmes. The regulation pathway of H2A.Z in diseases can be roughly summarized in two ways. Overexpression of H2A.Z in different types of diseases, especially in cancers, is one of important regulation pathways. The change of chromatin structure, recruitment of transcription factors, and coordination of other histone modifications are involved in the abnormal gene expression caused by H2A.Z overexpression in diseases. H2A.Z is upregulated and contributes to the disease-associated gene activation that drives pro-liferation, epithelial–mesenchymal transition (EMT), metastasis, and cancer in many diseases, such as hormone-dependent breast cancer, prostate cancer, liver cancer, hepatocellular carcinoma, melanoma, and uterine leiomyoma. H2A.Z silencing can inhibit the proliferation and invasion of intrahepatic cholangiocarcinoma (ICC) cells. On the other hand, the deposition of H2A.Z into the nucleosome or eviction from chromatin is more dependent on chromatin remodelers or histone chaperones. Deregulation of histone chaperone complexes can contribute to tumor development, and both its chaperone complexes, SRCAP and p400–TIP60, are affected. Components of the SRCAP complex are upregulated in several tumors, such as ovarian tumors, breast tumors, thyroid tumors, prostate tumors, melanoma, bladder tumors glioma, and Floating-Harbor syndrome (FHS). Furthermore, the chromatin remodeler p400 is overexpressed in some cancers, and promotes oncogenic WNT signaling and counteracts the tumor-suppressive function of the histone acetyltransferase KAT5. An aberrant nucleosome assembly has occurred, while excessive H2A.Z deposition in genome is mediated by H2A.Z overexpression and chromatin remodelers in various diseases. The nucleosome assembly dynamics containing H2A.Z, especially the competitive dynamics of H2A.Z and H2A in nucleosome assembly, has an important implication for enhancing the understanding of the role of H2A.Z in disease.

Reviewer 2 Report
Comments and Suggestions for Authors
In this manuscript, the authors examined the efficiency of nucleosome assembly and the stability of nucleosomes containing histone variant H2A.Z. The mathematical model of nucleosome assembly, derived from canonical histones, was applied to the system involving nucleosome assembly containing histone H2A.Z. The authors demonstrated that H2A.Z nucleosome assembly correlated with histone concentration, reaction rate constants, and reaction time. Furthermore, they demonstrated that canonical histones were assembled into nucleosomes more efficiently than H2A.Z-containing histones. Nucleosome heat denaturation assays revealed that H2A.Z-containing nucleosomes showed more stable than canonical nucleosomes. Although the topic of the manuscript is interesting and important, the paper requires some modifications before acceptance. Below are comments that should be addressed adequately.
In Figure 1, two lanes were represented for (A), (B), (D), and (E), and 6 lanes were represented for (C). What are the differences between two or six lanes?
In Figure 1C, is Slop Coefficient ‘Slope Coefficient’?
Although I do not understand the mathematical model of nucleosome assembly, it would be easier for readers to briefly explain how equation (2) can be derived from equation (1).
In Figure 2D, the authors performed time-course experiments. In Material and Methods, it was mentioned that nucleosome assembly reaction was performed with dialysis for 16 h with the buffer containing 2 M to 0.6 M salt (this was not clear to me), followed by further dialysis for an additional minimum of 3 h with TE. Which step was changed for this time course experiments?
In Figure 2B, the graphs seemed to converge to the same value of nucleosome assembly efficiently at 5 μg/mL of histone octamer. The equations shown in the graphs do not have intercepts. Does this mean that the authors gave 0 of nucleosome assembly efficiency at 0 μg/mL of histone octamers to obtain slopes? As the differences in slopes obtained from linear relationships depending on different incubation time were very small、 the calculation methods should be carefully described.
In Figure 3, the authors examined the effects of different DNA sequences on nucleosome assembly efficiency. The authors explained the differences between CS1, CS2, CS3, Cs4, CS5, and CS6 in their previous work. It would be easier for readers that these points should be mentioned more precisely in this manuscript (Figure legend or main text).
In Figure 3, how are the slopes of the plots for each graph calculated? Do the authors give 0 as intercepts?
In Figure 4, the efficiency of nucleosome assembly with histones containing H2A and H2A.Z were competitively analyzed. Western blotting method with peroxidase was generally very difficult. When input histone mixtures with different H2A/H2A.Z ratios were loaded on the same SDS-PAGE/western blotting, were both H2A and H2A.Z linearly detected?
In the materials and methods section 4.2, it was mentioned that nucleosome assembly reaction was performed with 3 ug DNA in 60 mL reaction volume. Is it correct? The concentration of DNA is calculated to be 50 pg/μL.
The subtitles for sections 4.3 and 4.4 should be changed.
Comments on the Quality of English LanguageI have no specific comments on English writing.
Author Response
International Journal of Molecular Sciences
Competitive chemical reaction kinetic model of nucleosome assembly using histone variant H2A. Z and H2A in vitro
Manuscript ID: ijms-2664452
Authors: Hongyu Zhao, Xueqin Shao, Mingxin Guo, Yongqiang Xing, Jingyan Wang, Liaofu Luo, Lu Cai
Dear reviewers and Ms. Aisa Safaya,
Thanks to the reviewers and editor for your helpful comments on our manuscript. We really appreciate that you give us the opportunity to revise the manuscript. We carefully revised the manuscript with consideration of reviewers’ comments and detailed corrections are listed below point by point.
Besides those changes made according to the reviewers’ suggestions, some minor revisions that will not influence the content and framework of the manuscript were also made in order to submit the manuscript more clearly. We do not list these minor revisions here. All changes in the revised manuscript are highlighted in yellow.
We hope this will make it acceptable for publication.
Sincerely yours,
Lu Cai (Email: nmcailu@163.com)
Oct.18, 2023
Reply to the reviewer 2:
Comment 1: In Figure 1, two lanes were represented for (A), (B), (D), and (E), and 6 lanes were represented for (C). What are the differences between two or six lanes?
Reply: Two repeated experimental DNAs or proteins were detected in two lanes for Figure 1(A), (B), (D), and (E). After native-PAGE, two repeated 601 DNA sequences were detected by ethidium bromide staining and Cy3-labeled fluorescence signal in Figure 1(A) and 1(B), respectively. Two repeated canonical histone octamers and H2A.Z-containing histone octamers were detected by SDS-PAGE in Figure 1(D) and 1(E), respectively.
In Figure 1(C), we loaded six different DNA sequences, referred to as CS1 to CS6, in six lanes to confirm the Cy3-labeled DNA sequences. The figure 1(C) was not clearly marked in the original manuscript, we have revised the annotations and legends in this figure.
Comment 2: In Figure 1C, is Slop Coefficient ‘Slope Coefficient’?
Reply: This is a mistake in the manuscript. The ordinate of Figure 2C has been changed to ‘Slope Coefficient’ in the revised version.
Comment 3: Although I do not understand the mathematical model of nucleosome assembly, it would be easier for readers to briefly explain how equation (2) can be derived from equation (1).
Reply: This is a helpful suggestion. We have added a briefly explanation for readers to understand the equation in our model.
In our previous work, we proposed a chemical kinetic model that provides a good description of canonical nucleosome reconstitution in vitro. Based on the chemical reaction kinetic principle, we obtained
|
(1) |
which gives a rigorous expression of N/S depending on the total concentration of DNA and histone octamer, reaction rate constant, and reaction time, where γ is the integral median of (k'/k). N, Q, and S indicate the nucleosome, total proteins, and total DNAs in the rection system, respectively. and T represent the mean reaction constant and total dialysis reaction time.
As γ/|Q-S| ≪1, Eq (1) can lead to following Eq (2).
(2)
Under , the exponential function in eq(2)is expanded to 2nd order of , and a simplified form of eq (2) for not-too-large Q is the linear relation between N/S and Q
|
(3) |
which is useful in analyzing experimental data.
Equation (3) provides a simplified linear relation between assembly efficiency of nucleosome and histone concentration, reaction rate constant, and reaction time for the following analysis of experimental data. In equation (3), T represent the mean reaction constant and total dialysis reaction time which can be control by the dialysis times of first step. The parameter N and S denote nucleosome DNA and total DNA, respectively, which can be detected by the fluorescence signal of Cy3-labeling DNAs. N/S can be used to assess nucleosome assembly efficiency. In the following, we will further test whether the kinetic model can be used to describe H2A.Z-containing nucleosome reconstitution.
Comment 4: In Figure 2D, the authors performed time-course experiments. In Material and Methods, it was mentioned that nucleosome assembly reaction was performed with dialysis for 16 h with the buffer containing 2 M to 0.6 M salt (this was not clear to me), followed by further dialysis for an additional minimum of 3 h with TE. Which step was changed for this time course experiments?
Reply: In the manuscript, we didn’t explain clearly the process of the experiment. For the time-course experiments of nucleosome assembly, the times of first-step dialysis with the buffer containing 2 M to 0.6 M salt were set at 10 h, 12 h, 14 h, and 16 h, respectively. And the second dialysis for an additional minimum of 3 h in TE buffer without NaCl was followed for all assembly experiments. This explanation has been added to section 4.2 in Material and Methods of revised manuscript.
Comment 5: In Figure 2B, the graphs seemed to converge to the same value of nucleosome assembly efficiently at 5 μg/mL of histone octamer. The equations shown in the graphs do not have intercepts. Does this mean that the authors gave 0 of nucleosome assembly efficiency at 0 μg/mL of histone octamers to obtain slopes? As the differences in slopes obtained from linear relationships depending on different incubation time were very small, the calculation methods should be carefully described.
Reply: In fact, the values of nucleosome assembly efficiencies under four assembly times at 5 μg/mL of histone octamer have very small differences in Figure 2(B). In accordance with the formulas, we set 0 of the nucleosome assembly efficiency at 0 μg/mL of histone octamers to produce slopes. In the revised manuscript, we provided a detailed description of the calculation methods.
Comment 6: In Figure 3, the authors examined the effects of different DNA sequences on nucleosome assembly efficiency. The authors explained the differences between CS1, CS2, CS3, Cs4, CS5, and CS6 in their previous work. It would be easier for readers that these points should be mentioned more precisely in this manuscript (Figure legend or main text).
Reply: This is a good suggestion for the improvement of manuscript. We have added the explanation of six CS sequences in the main text as follows.
CS1 to CS6 are six artificially designed DNA sequences with different sequence features to test the ability of nucleosome assembly in vitro. The length of the CS1-CS6 DNAs is 156 base pairs. As described in our previous report, CS1, CS2 and CS3 fit in with the nucleosome positioning pattern RRRRRYYYYY (R5Y5, R=Purine and Y=Pyrimidine) and contain uninterrupted eleven units of R5Y5. For the control group, CS4, CS5 and CS6 sequences consist of uninterrupted eleven copies of RRYRRYYRYY motif, RRYYRYYRRY motif and RYRYRYRYRY motif, respectively. In addition, contrary to the CS1 sequence, the CS2, CS3, CS4, CS5 and CS6 are characterized by visible 10.5-base periodicity of TA dinucleotides.
Comment 7: In Figure 3, how are the slopes of the plots for each graph calculated? Do the authors give 0 as intercepts?
Reply: The computational method of the slopes of the plots for each graph in Figure 3 was described in the section 4.3. While the liner relation between N/S and Q was fitted, we give 0 as intercepts for the following analysis of consistence with Equation.
Comment 8: In Figure 4, the efficiency of nucleosome assembly with histones containing H2A and H2A.Z were competitively analyzed. Western blotting method with peroxidase was generally very difficult. When input histone mixtures with different H2A/H2A.Z ratios were loaded on the same SDS-PAGE/western blotting, were both H2A and H2A.Z linearly detected?
Reply: Native-PAGE/Western blotting was employed to detect the efficiency of nucleosome assembly with histones containing H2A and H2A.Z. As described in section 4.2, the mixed nucleosomes were reconstituted on canonical histone octamer and H2A.Z-containing histone octamer. After confirmation of reconstituted nucleosome, the antibodies of H2A, H2A.Z, and H4 were used to detect the specific histones in re-constituted nucleosome, respectively.
In order to detect the specificity of antibodies, we set negative controls with 0 μg of histone H2A or histone H2A.Z in the input of nucleosome reconstituted reaction system, respectively. As expected, there is no any immunoblot in the negative bands, which indicated that the antibodies have the good immunological specificity. On the other hand, to reduce the bias caused by antibody recognition intensity among different antibodies, the quantized values of band at the maximum input of histone H2A or H2A.Z in reaction system was defined as 1, and the relative quantized values of the protein levels were standardized based on the maximum in each Western blotting assay, respectively. For example, for the competing reaction system with total histones of 3 μg, the reaction system of gradient 1 in table 1 contains protein H2A of 0 μg and protein H2A.Z of 3 μg. While this reaction system will be set as the negative control of histone H2A, the quantized values of this band was defined as 1 in the Western blotting assay with anti-histone H2A.Z antibody. Due to the normalization, when input histone mixtures with different H2A/H2A.Z ratios were loaded on the same native-PAGE/western blotting, both H2A and H2A.Z can be linearly detected.
The detailed methods are as follows. And this explanation has been added to the revised manuscript.
Firstly, native-PAGE separation of the samples in the H2A.Z and H2A competing nucleosome assembly systems was used to confirm the reconstituted nucleosomes. The Cy3 fluorescence of nucleosome DNA and the free DNA band were observed at an emission wavelength of 605 nm and an excitation wavelength of 520 nm.
Three same native-PAGEs were performed for the detection of H2A, H2A.Z, and H4 in nucleosome, respectively. The reconstituted samples containing equal mass proteins and DNAs were used for the electrophoretic loading. After native-PAGE separation, the histones in the nucleosome were transferred to the polyvinylidene difluoride (PVDF) membranes, respectively. The membranes were blocked with 5% skim milk at room temperature in a TBST buffer, and the indicated antibodies (H2A: Abcam, ab18255; H2A.Z: Abcam, ab4174; H4: Abcam, ab31830) were used, respectively. The immunoreactive bands were visualized using horseradish-peroxidase–conjugated secondary antibodies and an enhanced chemiluminescent substrate. Protein levels were quantified using Image J software. To reduce the experimental errors, the quantized values of band at the maximum input of histone H2A or H2A.Z in reaction system was defined as 1, and the relative quantized values of the protein levels by the free ImageJ software were standardized based on the maximum in each Western blotting assay, respectively. Via linear fitting between the and , it can be obtained the value of slope , while the vertical intercept was set to zero.
Comment 9: In the materials and methods section 4.2, it was mentioned that nucleosome assembly reaction was performed with 3 ug DNA in 60 mL reaction volume. Is it correct? The concentration of DNA is calculated to be 50 pg/μL.
Reply: This is a mistake in the manuscript. In the reaction system, 3 μg DNA templates in a total of 60 μL reaction volume were used to assemble nucleosomes. The concentration of DNA is calculated to be 50 ng/μL.
Comment 10: The subtitles for sections 4.3 and 4.4 should be changed.
Reply: “Native-PAGE/Western blotting analysis of competitive ability of histone H2A and H2A.Z in reconstituted nucleosomes” was used as the subtitle of section 4.4 in the revised manuscript.

Round 2
Reviewer 2 Report
Comments and Suggestions for Authors
I have looked at the revised manuscript and the author's responses to the comments from my previous review. The authors addressed all the points requested by this reviewer and I have no specific points to be modified. Although it is not essential, I think it would be better to omit unnecessary discussion, as the Discussion is too long.